# CCNLab: A Benchmarking Framework for Computational Cognitive Neuroscience

**Nikhil X. Bhattasali**
Stanford University
nikhilxb@cs.stanford.edu

**Momchil S. Tomov**
Harvard University
mtomov@g.harvard.edu

**Samuel J. Gershman**
Harvard University
gershman@fas.harvard.edu

## Abstract

CCNLab is a benchmark for evaluating computational cognitive neuroscience models on empirical data. As a starting point, its focus is classical conditioning, which studies how animals predict reward and punishment in the environment. CCNLab includes a collection of simulations of seminal experiments expressed under a common API, as wells as tools for visualizing and comparing simulated data with empirical data. CCNLab is broad, incorporating representative experiments from different categories of phenomena; flexible, allowing the straightforward addition of new experiments; and easy-to-use, so researchers can focus on developing better models. We envision CCNLab as a testbed for unifying computational theories of learning in the brain. We also hope that it can broadly accelerate neuroscience research and facilitate interaction between the fields of neuroscience, psychology, and artificial intelligence.

## 1 Introduction

Brains are the *de facto* standard for general intelligence [Lake et al., 2017], and many researchers believe that progress in artificial intelligence is intimately intertwined with understanding natural intelligence [Hassabis et al., 2017]. Modern research in neuroscience and psychology increasingly relies on computational models to express theories about how the brain works [Durstewitz et al., 2016, Jonas and Kording, 2017, Linderman and Gershman, 2017, Kriegeskorte and Douglas, 2018, Levenstein et al., 2020, Gershman, 2021]. This has brought the fields studying artificial and natural intelligence even closer, with computational neuroscientists directly borrowing ideas from machine learning [Montague et al., 1996, Yamins et al., 2014, Stachenfeld et al., 2017, Ma and Peters, 2020, Saxe et al., 2020] and vice versa [LeCun et al., 1995, He et al., 2016, Sutton and Barto, 2018]. This perpetuates a "virtuous cycle" in which the science and engineering of intelligence may progress together [Hassabis et al., 2017].

Despite these promising developments, there is a significant rift between machine learning and computational neuroscience in how research is evaluated and compared. Machine learning has benefited from the widespread adoption of publicly available datasets and benchmarks, like ImageNet [Deng et al., 2009] and OpenAI Gym [Brockman et al., 2016]. This has been instrumental for algorithm design, as the strengths and weaknesses of different work can be directly compared, and research is incentivized to perform beyond narrow, domain-specific tasks. In contrast, models in computational neuroscience have often been confined to a small set of phenomena in a specific domain, and due the lack of standards for evaluation, even those models can be difficult to compare.

35th Conference on Neural Information Processing Systems (NeurIPS 2021) Track on Datasets and Benchmarks.

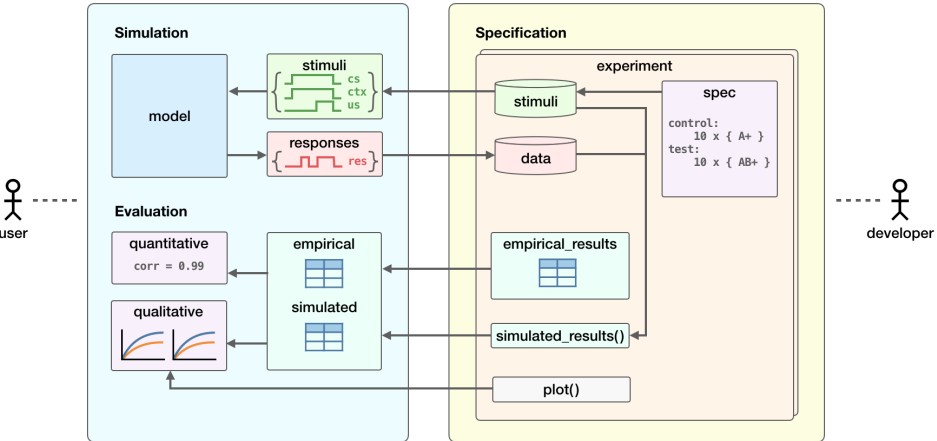

Figure 1: **Overview of CCNLab architecture.** Users can simulate their models on a wide variety of classical conditioning experiments, and the simulated results are evaluated against empirical results from published work. Each experiment is an environment that simulates a schedule of stimuli mirroring a real-world study.

We present **CCNLab** (short for Cognitive Computational Neuroscience Lab), a benchmark for computational models of classical conditioning. Using our Python framework inspired by OpenAI Gym [Brockman et al., 2016], users can simulate their models on a wide variety of classical conditioning experiments, and the simulated results are evaluated against empirical results from published work. Currently, the CCNLab registry includes 30 experiments to simulate the most established classical conditioning phenomena selected from the list provided in Alonso and Schmajuk [2012]. Our framework is modular and extensible, allowing developers to easily extend the benchmark with additional experiments from the literature or of their own design.

Classical conditioning has a rich history dating back to Pavlov [1927] and exhibits a diverse set of phenomena that have been extensively studied [Alonso and Schmajuk, 2012]. It has attracted much interest for computational modeling [Ertugrul and Tagluk, 2015, Gershman, 2015, Kutlu and Schmajuk, 2012b], and is closely related to reinforcement learning and optimal sequential decision making [Niv, 2009]. These factors make it an excellent starting point for introducing benchmarks into computational neuroscience and encouraging reproducible, rigorous evaluation of models. To that end, we perform a series of baseline experiments to evaluate existing methods on this benchmark. Our analysis reveals that, while our baselines are able to reproduce certain effects well, they lack the ability to generalize to a wider range of phenomena. We conclude by suggesting future extensions and research directions for computational modeling guided by publicly available benchmarks.

Our code for the benchmarks and baseline experiments has been open-sourced under the MIT License and is available at: https://github.com/nikhilxb/ccnlab.

## 2 Background

### 2.1 Classical Conditioning

A critical feature of intelligence is the ability to adapt in response to positive and negative feedback. The branch of machine learning that studies how agents can take actions that maximize reward is known as reinforcement learning [Sutton and Barto, 2018]. In neuroscience and psychology, learning based on rewards and punishments is studied through the paradigms of classical (Pavlovian) conditioning and instrumental (operant) conditioning. Classical conditioning studies how animals learn to predict outcomes in their environment, while instrumental conditioning studies how animals learn to select actions that lead to better outcomes.

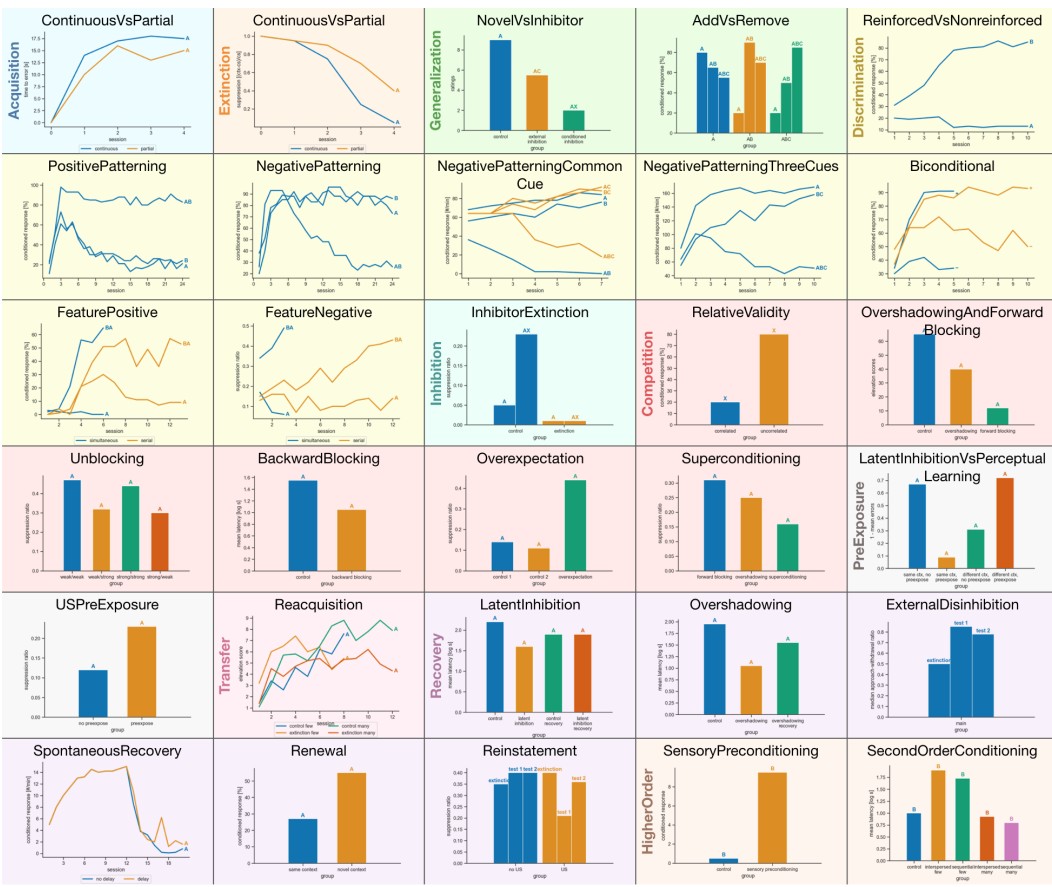

Figure 2: **Summary of simulated experiments.** The registry currently includes 30 experiments to simulate the most established classical conditioning phenomena selected from the list provided in Alonso and Schmajuk [2012] (see Appendix A for details). In the plots, experimental groups are separated by color, and stimuli are separated into individual bars/lines labeled with stimulus names.

In a typical classical conditioning experiment, the subject is presented with one or more conditioned stimuli (CS, e.g., a tone) followed by an unconditioned stimulus (US, e.g., food), which elicits an unconditioned response (UR, e.g., salivating) prior to training. After repeated pairing of the CS and the US over the course of multiple trials, the subject may begin to respond to the CS with what is known as the conditioned response (CR, e.g., salivating), as in the classic example of Pavlov's dog salivating in response to the sound of the bell [Pavlov, 1927]. The CR is a standard measure in classical conditioning experiments, and different schedules of CS stimuli result in different CR behavior, producing the wide range of classical conditioning phenomena that has been extensively cataloged [Alonso and Schmajuk, 2012].

## 2.2 Computational Modeling of Classical Conditioning

Reinforcement learning is deeply rooted in the study of conditioned behavior [Niv, 2009], dating back to Thorndike's seminal law of effect [Thorndike, 1898] and Skinner's principle of reinforcement [Skinner, 1935] in the early 20th century. By midcentury, the first formal treatments of animal learning were beginning to emerge in the field of mathematical psychology [Bush and Mosteller, 1951], paving the way for the famous Rescorla-Wagner model [Rescorla, 1972]. The Rescorla-Wagner model provided a formal account of a wide number of puzzling phenomenon in classical conditioning, such as blocking (the phenomenon where predictable rewards are poor reinforcers) and overshadowing (the phenomenon where salient stimuli tend to form stronger associations). It was this line of research that inspired the computer scientists Sutton and Barto to develop the foundational concepts of modern reinforcement learning theory [Sutton and Barto, 2018]. In the following decades,

a wealth of behavioral and neural evidence has further cemented the links between reinforcement learning theory and animal learning [Niv, 2009], most notably the discovery that dopamine neurons in the mammalian midbrain convey signals that closely correspond to the reward prediction errors prescribed by reinforcement learning theory [Schultz et al., 1997]. These findings have in turn spurred further research in reinforcement learning [Hassabis et al., 2017], leading to some remarkable recent success of reinforcement learning algorithms across a wide number of domains, from board games and video games [Mnih et al., 2015, Silver et al., 2017, Schrittwieser et al., 2020] to robotics [Haarnoja et al., 2018, Akkaya et al., 2019] to self-driving cars [Toromanoff et al., 2020].

Following the Rescorla-Wagner model, many other computational models of classical conditioning have emerged to explain a additional phenomena: the Pearce-Hall model posits that surprising outcomes increase learning rates [Pearce and Hall, 1980]; Temporal Difference models account for the temporal structure of neural learning signals [Schultz et al., 1997]; the Kalman Filter casts learning as Bayesian inference and accounts for effects of outcome uncertainty [Kakade, 2001]. More recently, Temporal Difference models operating on belief states have accounted for dopamine firing in the face of partial observability [Starkweather et al., 2017], and distributional reinforcement learning has accounted for the diversity of dopamine firing in the face of different reward distributions [Dabney et al., 2020]. Notably, these models account for overlapping yet distinct sets of phenomena. Despite promising recent work combining aspects of these models [Kutlu and Schmajuk, 2012b, Gershman, 2015], a unifying computational theory of animal learning is still lacking.

### 2.3 Benchmarks in Computational Neuroscience

In machine learning, it has long been recognized that standardized benchmarks and datasets are essential to the development of algorithms [Deng et al., 2009, Brockman et al., 2016]. However, in psychology and neuroscience, to the best of our knowledge there still do not exist standard benchmarks for comparing theories of animal learning. Alonso and Schmajuk [2012] provides a list of well-established classical conditioning phenomena. Subsequent work like Kutlu and Schmajuk [2012b] has sought to evaluate models based on this list. However, a public benchmark encoding these phenomena has not until now been created.

## 3 Technical Details

CCNLab provides a unified framework for specifying and simulating classical conditioning experiments. An **experiment** in CCNLab is analogous to an RL-style environment, as it simulates the interaction between the agent model and the world. Specifically, each experiment simulates the schedule of stimuli presented in a real-world classical conditioning experiment drawn from peer-reviewed academic research, and enables evaluation of the model's **simulated results** with the published **empirical results**.

Currently, the CCNLab registry includes 30 experiments (Figure 2) to simulate the most established classical conditioning phenomena selected from the list in Alonso and Schmajuk [2012]. These phenomena have been collected over a diverse range of conditioning preparations (e.g., visual/auditory conditioning, taste aversion, eyeblink conditioning, fear conditioning). Across experiments, the CSs were shapes, sounds, lights, flavors, and odors; the USs were food, shock, and mechanical stimulation; and both CSs and USs covered a wide range of intensities and durations. The inter-stimulus intervals (ISI) ranged from seconds to minutes; and the inter-trial intervals (ITI) ranged from seconds to days.

An overview of CCNLab architecture is provided in Figure 1. In the following sections, we describe the technical details of CCNLab, separating information most relevant for **users** (who are primarily concerned with evaluating the performance of their models in experiment simulations) and **developers** (who are interested in extending the benchmark with additional experiments).

### 3.1 Simulation and Evaluation

Users may run their models on experiments using the provided Jupyter notebooks [Kluyver et al., 2016] or custom Python scripts. Each experiment object encapsulates functionality for: (1) generating the stimuli that serve as model inputs, (2) storing model outputs (i.e., conditioned responses), (3) summarizing model outputs in a format that is directly comparable with empirical results gathered from published work, and (4) plotting simulated and empirical results. Here is an example of the API:

```python
import ccnlab.benchmarks.classical as classical
import ccnlab.evaluation as evaluation

# Select experiments to run, filtering by name using glob syntax.
for exp in classical.registry('*'):

  # Experiments often have multiple groups, each shown different stimuli.
  for g, group in exp.stimuli.items():

    # Users are free to decide how to initialize their models and how many
    # instances to allocate per group.
    for instance in range(N):
      model = YourModelHere()

      # Each group is shown a sequence of trials. At each timestep in a
      # trial, the model input consists of the conditioned stimuli (cs),
      # context (ctx), and unconditioned stimulus (us); the model output
      # is a response value.
      for i, trial in enumerate(group):
        for t, timestep in enumerate(trial):
          cs, ctx, us = timestep
          response = model.act(cs, ctx, us)
          exp.data[g][i][t]['response'].append(response)

  # Simulation results can be compared to empirical results from published
  # work, via evaluation metrics (quantitative) and plots (qualitative).
  empirical = exp.empirical_results
  simulated = exp.simulated_results()
  score = evaluation.correlation(empirical, simulated)
  exp.plot()
```

**Input Representation**     At each timestep in a trial, the environment provides:

1. `cs`: A list of active stimuli (string ids) and their magnitudes (positive real-valued). For most experiments, the magnitudes are either 0 or 1.

2. `ctx`: The active context (string id). The context remains the same throughout a trial.

3. `us`: The unconditioned stimulus magnitude (positive real-valued). For most experiments, the magnitude is either 0 or 1.

Alternatively, the `cs` and `ctx` are available as one-hot vectors with dimensions equal to the stimuli and context space, respectively. The 3 components of the observation are provided separately as each has unique semantic meaning. By definition, the US evokes a response in untrained subjects while the CS do not [Pavlov, 1927]. Moreover, multiple studies have demonstrated the importance of context for modulating conditioning effects to the CS [Alonso and Schmajuk, 2012, Gershman, 2017]. It is therefore common for models of classical conditioning to treat these input signals differently.

**Output Representation**     At each timestep in a trial, the model should provide a response value (real-valued) indicating the strength of the conditioned response. Due to the diversity of conditioning preparations used across experiments, a degree of standardization is needed to enable the same model to perform across preparations. In particular, following Kutlu and Schmajuk [2012b], the response value should represent the abstract CR itself, regardless of whether it is experimentally measured as an increase or decrease of some behavior. For example, in an appetitive preparation like eyeblink conditioning where the CR is measured as an increase in blinking activity, a lower response value represents *lower* blinking activity and a higher response value represents *higher* in blinking activity, as expected. Conversely, in an aversive preparation like bar pressing where the CR is measured as a reduction in bar pressing activity, a lower response value of represents *baseline/higher* bar pressing activity and a higher response value represents *reduced/lower* bar pressing activity.

**Experiment Parameters**  Each experiment simulates the schedule of stimuli presented in its real-world counterpart. Due to the diversity of conditioning preparations used across experiments, the number of conditioning trials vary significantly. By default, the experiments provide a number of trials in each phase proportional to the original work or the simulations performed in Kutlu and Schmajuk [2012b]. However, it is possible to change the number of trials per session, if desired.

**Evaluation Metrics**  We follow the approach of evaluating models using scale-invariant, ordinal measures of the quality of fit between empirical and simulated results. Following Kutlu and Schmajuk [2012b], we use Pearson's correlation coefficient $r$, which yields a value between -1 and 1 reflecting a linear correlation between empirical and simulated results. A special case is when the empirical results contain only 2 data points, in which case we use the ratio (smaller to larger) of ratios between the data points, which yields a value between 0 and 1.

## 3.2 Specification

Developers may extend the benchmark with additional experiments from the literature or of their own design. CCNLab provides a collection of data structures and functions to make it easy to implement experiments, including: (1) an abstract syntax library for specifying experimental stimuli, (2) processing functions for computing conditioned responses and suppression ratios from the raw data, and (3) plotting functions to generate line or bar graphs.

Here is an example of the API, which has been simplified for clarify (for more details, refer to the code repository):

```python
import pandas as pd
import ccnlab.benchmarks.classical.core as cc

@cc.registry.register
class Acquisition_ContinuousVsPartial(cc.ClassicalConditioningExperiment):
  def __init__(self, n=64, prob=0.5):
    # Specify stimuli structure for each experimental group using the
    # abstract syntax.
    super().__init__({
      'continuous': cc.seq(
          cc.trial('A+'),
          repeat=n, name='train'
        ),
      'partial': cc.seq(
          cc.sample({ cc.trial('A+'): prob, cc.trial('A-'): 1 - prob }),
          repeat=n, name='train',
        ),
    })

    # Encode empirical results and configure plotting.
    self.empirical_results = pd.DataFrame(
      columns=['group', 'session', 'A'],
      data=[ ... ]
    )
    self.plots = [ lambda df, ax: cc.plot_lines(df, ax=ax, x='session') ]

  # Transform raw model responses to the same format as empirical results.
  def simulated_results(self):
    df = self.dataframe(lambda x: {
      'A': cc.conditioned_response(x['timesteps'], x['response'], ['A']),
    })
    return cc.trials_to_sessions(df, self.trials_per_session)
```

**Abstract Syntax Library**  Each experiment presents a schedule of stimuli consisting of multiple trials per group. To facilitate the specification of stimuli structure, we developed an abstract syntax library that allows classical conditioning experiments to be expressed in a consistent way. By

composing a sequence of nodes, developers may specify the structure using an abstract syntax tree, which can then be compiled into the a sequence of trials, each consisting of multiple timesteps. The syntax closely conforms to standard classical conditioning notation, using the following nodes:

- `Stimulus`: Leaf node specifying the presentation of a stimulus (string), its magnitude (float), and its start and end timesteps in a trial (ints).
- `Trial`: Compound node specifying a trial consisting of the CS (list of `Stimulus`), CTX (string), and US (`Stimulus`);
- `Sample`: Compound node specifying probabilities with which to choose each of its children.
- `Sequence`: Compound node specifing a sequence of nodes, how many times to repeat the sequence, and a name (used for naming different phases of the schedule).

**Empirical Results**  The published work that the experiments simulate present their results in the form of summary statistics. Typically, the measure of interest is the conditioned response or a suppression ratio, and this measure is plotted over trials/sessions (line plots) or over groups (bar plots). For data collection, we relied on numerical tables in published work if available; otherwise, we digitally enlarged the plots and used a grid overlay to more precisely estimate the measure values.

**Simulated Results**  In order to compare simulated results to the empirical results, the raw model responses must first be transformed into the same summary statistics as the empirical results. We are given a set of active conditioned stimuli $\texttt{cs}_t$ and model responses $\texttt{response}_t$ for each timestep $0 \le t < T$ in a single trial.

For appetitive preparations, the conditioned response to a stimulus x is the average response during the presentation of x.

$$\texttt{ConditionedResponse}(\texttt{x}) = \frac{\sum_{t:\texttt{cs}_t=\texttt{x}} \texttt{response}_t}{\sum_{t:\texttt{cs}_t=\texttt{x}} 1}$$

For aversive preparations, the suppression ratio to a stimulus x is the ratio of (reduced) responding during the presentation of x compared to baseline responding. As described in Section 3.1, the abstract response value must be inverted to correspond to aversive CR behavior. We use the maximum response value for inversion as in Kutlu and Schmajuk [2012b].

$$\texttt{SuppressionRatio}(\texttt{x}) = \frac{\sum_{t:\texttt{cs}_t=\texttt{x}} (\max_t \texttt{response}_t) - \texttt{response}_t}{\sum_t (\max_t \texttt{response}_t) - \texttt{response}_t}$$

Finally, after computing the measure for each trial, the measures are aggregated across subjects through averaging. If necessary, consecutive spans of trials are aggregated into sessions through averaging.

## 4  Baselines and Experiments

To illustrate how the benchmark can be used in practice, we applied it to compare 3 classical conditioning baseline models from Gershman [2015]. For more details on the models and their parameters, see Appendix A.

- **Rescorla-Wagner** [Rescorla, 1972]: The predicted reward (US) is a linear combination of the input stimuli (CS), weighted by their associative weights. The weights are updated in proportion to the reward prediction error – the difference between the predicted and actual reward – and how active the stimuli are.
- **Kalman Filter** [Kakade, 2001]: A Bayesian extension of Rescorla-Wagner which learns the covariance of the weights in addition to their mean. The learning rate is dependent on the uncertainty encoded by the covariance.
- **Temporal Difference Learning** [Sutton, 1988]: A temporal extension of Rescorla-Wagner which predicts cumulative future reward instead of immediate reward only.

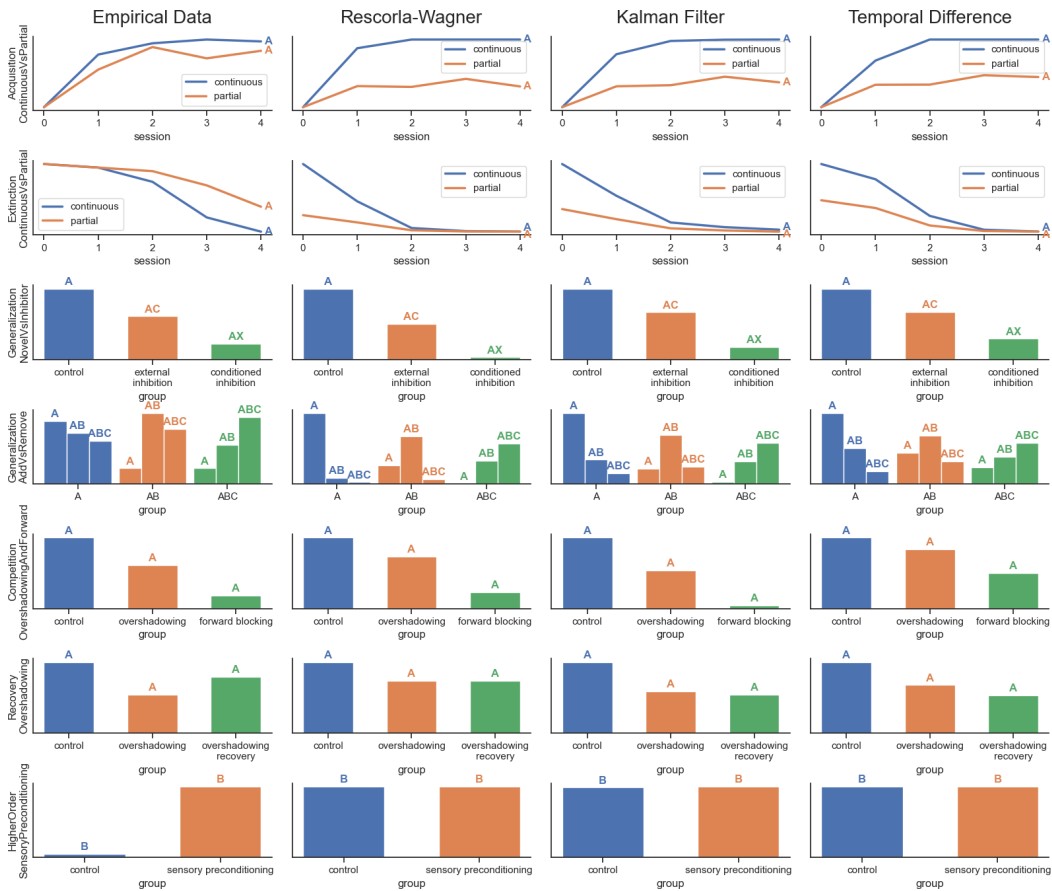

Figure 3: **Plots for selected experiments.** Models can be evaluated qualitatively by comparing plots for empirical and simulated results. Column 1: Empirical results from published work. Columns 2-4: Simulated results for each model. Plot coloring and notation are the same as Figure 2.

We evaluated the models based on how well they fit the empirical results for each experiment according to the metrics in Section 3.1. Scores for all experiments are presented in Table 1, and scores averaged across categories are presented in Table 2. We also plotted the empirical and simulated results, for which a subset of experiments are shown in Figure 3.

Unsurprisingly, Rescorla-Wagner is able to account for phenomena it was designed to explain, such as acquisition (Figure 3, Row 1), extinction (Row 2), external and conditioned inhibition (Row 3), generalization with added and removed cues (Row 4), and overshadowing and forward blocking (Row 5). Kalman Filter and Temporal Difference Learning are also able to account for these phenomena well, since they are generalizations of Rescorla-Wagner. In addition, they each account better for other phenomena, such as extinction of inhibition for Kalman Filter; and overexpectation, superconditioning, and higher-order conditioning for Temporal Difference Learning, giving them higher scores overall (Table 2). Notably, all models perform poorly on most experiments in the benchmark, and even show negative correlations on a substantial number of experiments (Table 1). This could be partially explained by the fact that model parameters were not tuned to each experiment, although we found that performance did not change substantially with different parameter settings (data not shown). We take this to highlight the limitations of these standard learning models which are still widely used in neuroscience, suggesting a pressing need for the development and adoption of more general theories of learning.

Table 1: **Scores for all experiments.** Models can be evaluated quantitatively by computing measures of fit between empirical and simulated summary statistics. These scores use the Pearson correlation, except for experiments indicated by * which use ratio of ratios instead (see Section 3.1). Scores greater than 0.8 are highlighted in bold.

| Category | Experiment | Rescorla-Wagner | Kalman Filter | Temporal Difference |
|---|---|---|---|---|
| Acquisition | `ContinuousVsPartial` | **0.80** | **0.83** | **0.85** |
| Extinction | `ContinuousVsPartial` | 0.54 | 0.57 | 0.69 |
| Generalization | `NovelVsInhibitor` | **1.00** | **0.99** | **1.00** |
| | `AddVsRemove` | 0.60 | 0.75 | 0.60 |
| Discrimination | `ReinforcedVsNonreinforced` | -0.88 | -0.88 | -0.89 |
| | `PositivePatterning` | -0.82 | **0.88** | **0.89** |
| | `NegativePatterning` | -0.64 | -0.63 | 0.75 |
| | `NegativePatterningCommonCue` | 0.15 | 0.02 | 0.74 |
| | `NegativePatterningThreeCues` | -0.76 | -0.70 | 0.46 |
| | `Biconditional` | 0.23 | 0.33 | 0.68 |
| | `FeaturePositive` | -0.30 | -0.07 | 0.11 |
| | `FeatureNegative` | 0.33 | 0.42 | 0.28 |
| Inhibition | `InhibitorExtinction` | -0.36 | **0.99** | 0.48 |
| Competition | `RelativeValidity*` | 0.00 | 0.00 | 0.00 |
| | `OvershadowingAndForwardBlocking` | **0.99** | **1.00** | **0.99** |
| | `Unblocking` | -0.65 | -0.65 | -0.17 |
| | `BackwardBlocking*` | 0.17 | 0.12 | 0.77 |
| | `Overexpectation` | -1.00 | -0.99 | **0.87** |
| | `Superconditioning` | -0.77 | -0.65 | **0.88** |
| PreExposure | `LatentInhibitionVsPerceptualLearning` | 0.00 | 0.00 | 0.00 |
| | `USPreExposure*` | **0.81** | 0.62 | **0.96** |
| Transfer | `Reacquisition` | 0.72 | 0.74 | 0.63 |
| Recovery | `LatentInhibition` | 0.00 | 0.01 | 0.00 |
| | `Overshadowing` | **0.83** | 0.78 | 0.62 |
| | `ExternalDisinhibition` | 0.69 | 0.58 | 0.40 |
| | `SpontaneousRecovery` | **0.97** | **0.93** | 0.56 |
| | `Renewal*` | 0.00 | 0.00 | 0.00 |
| | `Reinstatement` | -0.71 | -0.70 | -0.82 |
| HigherOrder | `SensoryPreconditioning*` | 0.00 | 0.00 | 0.05 |
| | `SecondOrderConditioning` | 0.01 | 0.17 | 0.49 |

Table 2: **Scores for all categories.** Averages across categories of the scores in Table 1. Scores greater than 0.8 are highlighted in bold.

| Category | Rescorla-Wagner | Kalman Filter | Temporal Difference |
|---|---|---|---|
| Acquisition | **0.80** | **0.83** | **0.85** |
| Extinction | 0.54 | 0.57 | 0.69 |
| Generalization | **0.80** | **0.87** | **0.80** |
| Discrimination | -0.34 | -0.08 | 0.38 |
| Inhibition | -0.36 | **0.99** | 0.48 |
| Competition | -0.21 | -0.20 | 0.56 |
| PreExposure | 0.41 | 0.31 | 0.48 |
| Transfer | 0.72 | 0.74 | 0.63 |
| Recovery | 0.30 | 0.27 | 0.13 |
| HigherOrder | 0.01 | 0.09 | 0.27 |
| Overall | 0.27 | 0.44 | 0.53 |

# 5    Conclusion and Future Work

We presented CCNLab, an open-source benchmark for computational modeling of classical conditioning. To the best of our knowledge, this is the first benchmark of its kind for evaluating and comparing computational models in neuroscience and psychology. We hope that it will encourage the development of broader neuroscientific theories of learning, as well as the development of powerful artificial intelligence algorithms inspired by the brain. In the following sections, we suggest future directions for the benchmark and for research using the benchmark.

**Benchmark Extensions**    While the presented benchmark is a step forward for the rigorous evaluation of computational classical conditioning models, there are a number of extensions that would improve it. (1) Currently, it contains a selection of experiments demonstrating the most well established phenomena. We expect the registry of experiments to grow over time with contributions from the authors and the community, adding experiments that capture a wider breadth of phenomena, especially regarding multi-modal combination and temporal effects that are presently missing. (2) There is significant diversity in the types of conditioning preparations reflected by the experiments. For more a standardized comparison, it would be useful to capture the phenomena in a single preparation, or show the same phenomena across multiple preparations. (3) Currently, we only allow a single US and UR, but the model should likely be extended handle multiple. (4) While current experiments test the fit of simulated and empirical behavioral data, it is ultimately important for models of classical conditioning to also capture neural data.

**Research Directions**    (1) From our experiments it is evident that computational models of classical conditioning still fail to explain a breadth of phenomena. We hope that the introduction of this benchmark will encourage future research to consider a broader range of phenomena. (2) We hope that CCNLab is only the first step towards widespread benchmarking in computational neuroscience. In the future, we expect that benchmarks for different domains will be added, for instance to capture the diverse phenomena of instrumental conditioning and short-term/working memory [Oberauer et al., 2018]. (3) Finally, we believe that it will continue to be productive for the fields of artificial intelligence and computational neuroscience to build on each other's insights, simultaneously approaching the project of intelligence from both a scientific and engineering standpoint.

## Acknowledgments and Disclosure of Funding

This work was supported by grants from the NIH (U19 NS113201-01), the Toyota Corporation, the Air Force Office of Scientific Research (FA9550-20-1-0413), and the Center for Brains, Minds and Machines (CBMM), funded by NSF STC award CCF-1231216. The funders had no role in study design, data collection and analysis, decision to publish or preparation of the manuscript. The authors have no competing interests to report.

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
