# OpenReview forum: "CCNLab: A Benchmarking Framework for Computational Cognitive Neuroscience"
_NeurIPS.cc/2021/Track/Datasets_and_Benchmarks/Round1 — NeurIPS 2021 Datasets and Benchmarks Track (Round 1)_

### Official Review · Reviewer_EzEv · 2021-07-01

**Rating:** 7
**Confidence:** 2

**Strengths:**

The paper proposes a benchmark of classical conditioning tasks to simplify comparison of computational models of conditioned behavior. It presents an analysis of about 30 experiments across about 10 types of conditioning tasks, with three classical conditioning models (Rescorla-Wagner, Kalman Filter, Temporal Difference Learning), and a pipeline to incorporate new models. This is an important work as it highlights the needs for benchmarks in computational neuroscience to encourage model development and standardized testing.


**Weaknesses:**

The paper is very well-written, but as a novice in this area, I found it a bit difficult to understand the terminology: please define classical conditioning on first use (in abstract). Please also move some of the technical details (e.g., the API in section 3.2), to supplementary material, and instead describe the tasks and experiments in more detail. Do the classical conditioning tasks presented cover a wide variety of tasks, and are they representative? In Figure 3, it is difficult to see what is the expected (published) result, and what is the result of 3 models considered? Please also define the notation directly in the caption or in the figure itself.

It would help to discuss whether the proposed models are representative of humans only, animals or other models of behavior.


**Additional Feedback:**

Thank you for clarifications!

**Clarity:**

The paper is well-written and the benchmark is well-documented.


**Correctness:**

The paper presents a benchmark, and some analysis is done in lines 223-232. It would really help to expand this section, adding quantitative measures, and explaining the conclusions, pointing to the findings in Figure 3.


**Documentation:**

The benchmark is hosted on Github with appropriate licensing. I have not tried to reproduce it, but it seems well-documented and complete.


**Relation To Prior Work:**

Related work is clearly discussed and easy to understand.


**Summary And Contributions:**

The paper proposed a benchmark of evaluating computational models on classical conditioning tasks (“how to learn to predict outcomes in their environment“, e.g., acquisition, generalization, recovery etc.), where simulations are directly compared to published results from other work.

---

> ### Author Response · Authors · 2021-07-12
> **Authors Response**
>
> Thank you for your helpful feedback!
>
> **Definitions** — As suggested, we’ve made changes to the abstract to define the terminology earlier on, so that it’s hopefully clearer to readers less familiar with the area.
>
> **Technical/experimental details** — We opted to keep the API in the main text to mirror to the format of similar work [Brockman et al. 2016, Tassa et al. 2018]. As suggested, we’ve added more information to the experiment descriptions (Appendix A) to detail the preparation info (animal, aversive/appetitive, etc.).
>
> **Experiment selection** — Indeed, we aimed to curate a broad and representative selection of experiments to simulate the most established classical conditioning phenomena from the extensive list of [Alonso and Schmajuk 2012]. Our experiments span various categories of phenomena (Acquisition, Generalization, etc.) as well as different conditioning preparations.
>
> **Figure notation** — We’ve updated Figure 2 and Figure 3 captions to clarify the notation used in the bar/line graphs.
>
> **Analysis** — We’ve expanded the analysis in Section 4 to compare models based on their scores across experiments and provide more general conclusions.

---

### Official Review · Reviewer_m9Ct · 2021-07-02
**An interesting set of benchmarks for future research**

**Rating:** 9
**Confidence:** 3
**Clarity:** Yes, the paper is clear.

**Strengths:**

**Significance:** Assuming the authors are correct, these benchmarks are new and unique to the field.

**Relevance:** The convergence of natural and artificial intelligence is highly relevant to the broader research community

**Accessibility and accountability:** The authors include a Jupyter notebook with examples.

**Weaknesses:**

- For the install, it would be helpful to list the dependencies (presumably python 3.7).
- How can researchers add their contributions? Please include a way for others to contribute in a formalized way


**Additional Feedback:**

n/a

**Correctness:**

Construction appears sound: The authors chose a number of experiments from the previous literature, and chose a few different more recent models for comparison.

**Documentation:**

Minor improvements in documentation (installation requirements), but the included Jupyter notebook is helpful. Perhaps more documentation on how to make and contribute an experiment would be welcome.

**Ethics:**

As these experiments are all derived from previous literature, it might be helpful to note the demographics and variability of the responses, lest the future trainings be based on a dataset without sufficient diversity.

**Relation To Prior Work:**

Yes, extensive previous work (where applicable) is included.

**Summary And Contributions:**

Here the authors present CCNLab, a benchmark for comparing computational neuroscience models, with seed examples of classical conditioning experiments. The authors clearly describe the background relationship between computational neuroscience research and machine learning, and suggest classical conditioning as one example of many that could be used to assist both artificial and natural intelligence researchers. Some minor requests included below.

---

> ### Author Response · Authors · 2021-07-12
> **Authors Response**
>
> Thank you for your helpful feedback!
>
> **Installation** — We’ve updated the installation docs and now include a conda environment to facilitate installing the necessary Python and pip dependencies.
>
> **Contributing** — We will add a CONTRIBUTING.md doc to detail the process for adding experiments to the benchmark.

---

### Official Review · Reviewer_HtZH · 2021-07-04
**Great initiative towards building a general computational framework for classical conditioning**

**Rating:** 8
**Confidence:** 4
**Correctness:** The claims are reasonable.
**Clarity:** Yes, the paper is written clearly.

**Strengths:**

1.	A considerably large collection of classical conditioning experiments with scope for additions in the future
2.	Easy to use API and simple evaluation metrics
3.	Will be useful for the broader research community (esp. with future additions of multiple CS scenarios and instrumental conditioning) and will provide a link between AI and neuroscience to tackle the questions of cognition and intelligence synergistically.

**Weaknesses:**

Since the authors have already charted out how this benchmark will be updated in the future by expanding its scope (both by adding experiments and including ways to model neural data), I will use this space to indicate a couple of things that could potentially increase the value of this benchmark for the community.
1.	The benchmark, as described in the manuscript, evaluates the model performance independently on each experiment. However, it will be useful to have an overall score for each model or model class (if the parameters are trained for each experiment) so that one can arbitrate amongst multiple competing models and pick the best model that explains the gamut of experimental observations (see BrainScore -- http://www.brain-score.org/). One such measure could be just an average of all the scores, but the authors can also consider other more sophisticated measures computed by weighting experiments differently (based on the strength of the experimental evidence?). In fact, there could be a summary score for each class of experiments.
2.	As the authors plan to expand the benchmark, it will be important to include the animal models these experiments were run on. It is not clear whether all experiments in the current benchmark were run on rodents (whether rats or mice?), and going forward it will be useful to have this information 1) to figure out potential differences in these experimental phenomena across species; and 2) to build computational models that could reveal convergent or divergent neural mechanisms across species.


**Additional Feedback:**

None

**Documentation:**

I was able to access the dataset hosted on GitHub, but did not test it to assess reproducibility.

**Ethics:**

Does not apply

**Relation To Prior Work:**

Prior work is cited appropriately and discussed in the context of the paper

**Summary And Contributions:**

The authors have created a benchmark for testing computational models of Classical Conditioning (CC) -- a widely studied phenomenon with a popular history dating back to Pavlov and his dog. Recently, classical conditioning has garnered much interest from the computational neuroscience and machine learning communities because of its relation to reinforcement learning. Given the importance of CC in cognition and an urgent need for a general computational framework, this submission contributes by establishing a benchmark with reproducible results from literature and an easy-to-use API to test computational models.

---

> ### Author Response · Authors · 2021-07-12
> **Authors Response**
>
> Thank you for your helpful feedback!
>
> **Aggregated score** — We agree that such measures will allow easier comparison of models, and have updated the manuscript to include an average score for each category of phenomena (Acquisition, Generalization, etc.). We thought that separating scores by category would allow more fine-grained and meaningful insights than a score that aggregates across all experiments, which would over-represent categories that have more implemented experiments.
>
> **Conditioning preparations** — This would indeed be useful information to include so that users don’t have to keep looking it up via the experiment citations. We’ve updated the code so that preparation info (animal, aversive/appetitive, etc.) is included via various standardized, searchable fields in each experiment’s metadata dict. We’ve also updated Appendix A to provide this info.

---

### Decision · Program_Chairs · 2021-07-27

**Decision:**

Accept

**Comment:**

The authors introduce a benchmark for testing computational cognitive neuroscience models. The reviewers noted the extensive scope of the benchmark and the quality of the API and there is a consensus on acceptance.